# Hybrid Beads of Zero Valent Iron Oxide Nanoparticles and Chitosan for Removal of Arsenic in Contaminated Water

Mian Fawaz Ahmed [1], Muhammad Asad Abbas [1], Azhar Mahmood [2], Nasir M. Ahmad [1,*], Hifza Rasheed [3], Muhammad Abdul Qadir [4], Asad Ullah Khan [5], Hazim Qiblawey [6], Shenmin Zhu [7], Rehan Sadiq [8] and Niaz Ali Khan [9,10,*]

[1] Polymer Research Lab, School of Chemical and Material Engineering, National University of Sciences and Technology (NUST), Islamabad 44000, Pakistan; mianfawaz@gmail.com (M.F.A.); engr.ma.abbas@live.com (M.A.A.)

[2] School of Natural Sciences, National University of Sciences and Technology (NUST), Islamabad 44000, Pakistan; dr.azhar@sns.nust.edu.pk

[3] National Water Quality Laboratory, Pakistan Council of Research in Water Resources, Islamabad 44000, Pakistan; pcrwr2005@yahoo.com

[4] The Institute of Chemistry, University of the Punjab, Quaid-e-Azam Campus, Lahore 54770, Pakistan; mabdulqadir.chem@pu.edu.pk

[5] Department of Chemical Engineering, COMSATS University Islamabad, Lahore Campus, Lahore 54000, Pakistan; asadkhan@cuilahore.edu.pk

[6] Department of Chemical Engineering, College of Engineering, Qatar University, Doha P.O. Box 2713, Qatar; hazim@qu.edu.qa

[7] State Key Laboratory of Metal Matrix Composites, Shanghai Jiao Tong University, Shanghai 200240, China; smzhu@sjtu.edu.cn

[8] Faculty of Applied Sciences, School of Engineering, University of British Colombia, Okanagan Campus, Kelowna, BC V1V 1V7, Canada; rehan.sadiq@ubc.ca

[9] School of Chemical Engineering and Technology, Tianjin University, Tianjin 300072, China

[10] Nova Materials Technologies, Lahore 54900, Pakistan

* Correspondence: nasir.ahmad@scme.nust.edu.pk (N.M.A.); niaz@tju.edu.cn (N.A.K.)

**Abstract:** Water contaminated with highly hazardous metals including arsenic (As) is one of the major challenges faced by mankind in the present day. To address this pressing issue, hybrid beads were synthesized with various concentrations of zero valent iron oxide nanoparticles, i.e., 20% (FeCh-20), 40% (FeCh-40) and 60% (FeCh-60) impregnated into a polymer of chitosan. These hybrid beads were employed as an adsorbent under the optimized conditions of pH and time to facilitate the efficient removal of hazardous arsenic by adsorption cum reduction processes. X-ray Diffraction (XRD), Scanning Electron Microscopy (SEM), Fourier Transform Infrared Spectroscopy (FTIR), Brunauer- Emmett-Teller BET, a porosity test and wettability analysis were performed to characterize these hybrid beads. The porosity and contact angle of the prepared hybrid beads decreased with an increase in nanoparticle concentration. The effects of various adsorption factors such as adsorbent composition, contact period, pH value and the initial adsorbate concentration were also evaluated to study the performance of these beads for arsenic treatment in contaminated water. FeCh-20, FeCh-40 and FeCh-60 have demonstrated 63%, 81% and 70% removal of arsenic at optimized conditions of pH 7.4 in 10 h, respectively. Higher adsorption of arsenic by FeCh-40 is attributed to its optimal porosity, hydrophilicity and the presence of appropriate nanoparticle contents. The Langmuir adsorption kinetics described the pseudo second order. Thus, the novel beads of FeCh-40 developed in this work are a potent candidate for the treatment of polluted water contaminated with highly toxic arsenic metals.

**Keywords:** zero valent iron oxide; chitosan; nanoparticles; arsenic; adsorption; isotherm model; kinetics

## 1. Introduction

Contamination of water bodies is one of the crucial hazardous factors affecting the well-being of our earth and its inhabitants. The environment, economic growth, and development are sacrificially affected by a sub-standard quality of surface and ground water. Contamination of water by toxic elements such as Cr, Pb, Hg, Cd and As, etc., is among the major threats to the health of human beings as well as aquatic and terrestrial life [1]. Generally, heavy metals are water soluble, and hence, have facile mobilization in the environment. Arsenic is one of the elemental pollutants that has caused significant harm to aquatic and human health. As per World Health Organization (WHO) guidelines, 10 µg/L is a permissible arsenic limit for potable water [2]. Arsenic-contaminated drinking water can cause various health problems such as skin lesions, lung carcinoma, and cardiovascular ailments. Worldwide, about 150 million people are affected by arsenic-contaminated aquifers. In Pakistan, the situation is even more critical due to high pH dissolution of arsenic from basic surface soil into Arid Indus areas and substantial irrigation by these contaminated aquifers, thus posing a threat to the health of 50–60 million people using subterranean water in this region [3]. These figures are distressing and call for appropriate mitigation measures.

Various conventional methods are employed for arsenic treatment in water including adsorption [4], extraction [5], reverse osmosis, oxidation [6], electrodeposition [7], ion exchange resin and membrane screening [8]. There is room for improvement in terms of duration, effectiveness, adsorption capacity and cost to broaden the scope of these techniques for water treatment applications [9]. Recently, the adsorption technique has become the most promising wastewater treatment method, being economical, environment friendly, easy to operate, flexible and devoid of by-products as well as having the possibility of regenerating the adsorbent for reuse up to a feasible number of cycles [10,11]. However, the use of adsorbents for pollutant removal has its challenges, such as the limited number of adsorbent materials available for column configuration and their non-suitability for different types of pollutant in wastewater. These shortcomings can be minimized by employing macromolecules and nanoparticles capable of deep particle adsorption, thus increasing the diffusion rate by increasing the surface area of the adsorbent [12]. Various nanoparticles have been used as adsorbents such as activated carbon, silver [13], aluminum oxide, titanium oxide [14], polymer and metal oxide [15]. Due to the high affinity and selectivity of iron oxide nanoparticles toward arsenic, they are considered highly efficient for the adsorption of arsenic. However, the inadequate fabrication of iron oxide into a bead structure limits its application in the column adsorbent bed. Combining iron oxide with environmentally benign functional biopolymers may address this challenge. Chitosan is a carbohydrate polymer that has a benign nature, natural occurrence, is nontoxic, biocompatible, biodegradable, environmentally friendly, and feasible to process into films and beads. Moreover, it is able to form a complex with heavy metals in water owing to the presence of amino and hydroxyl groups [16,17]. However, it faces some challenges such as high swelling behavior and nonporous texture that contribute towards a reduction in adsorption capacity. This can be overcome by developing a hybrid material through the incorporation of NPs impregnated into chitosan [14,18].

Herein, we report on combining the above-mentioned advantages of nano zero valent iron oxide particles (NZVI) and chitosan to develop hybrid beads of optimum composition for effective arsenic treatment in contaminated water [19]. NZVI particles impregnated into the chitosan matrix synergistically increased adsorbent efficiency for arsenic in aqueous media. These hybrid beads were characterized to study their chemical structures, morphology, porosity, and surface wettability [20]. The performance of the hybrid beads was measured with respect to various parameters such as initial arsenic concentration, time and pH. Various theorems such as pseudo first order, pseudo second order, Langmuir and Freundlich isotherms were applied to study the adsorption mechanism.

## 2. Experimental Work

### 2.1. Materials

Iron chloride hexa-hydride (FeCl$_3$.6H$_2$O) (Sigma-Aldrich, Mw = 270.30 g/mol), chitosan Mw = $\left(4.9 \times 10^5\right)$ (Sigma-Aldrich), ethylene glycol (Sigma-Aldrich, Germany, Mw = 62 g/mol), and sodium acetate (Sigma-Aldrich, Germany, Mw = 82 g/mol) were obtained commercially. Different aqueous solutions of Arsenic (V) were made by dissolving pre-calculated amounts of arsenic (As$_2$O$_5$, Sigma-Aldrich, Mw = 229.8 g/mol) into ultra-pure water according to ASTM standards (D2972-08), while the pH of the medium was modified by adding the required concentration of a NaOH and HCl solution. All reagents and solvents used in this study were of analytical grade and used without further treatment unless specified.

### 2.2. Synthesis of Iron Oxide Nanoparticles

Iron oxide nanoparticles were prepared by a hydrothermal method [21]. Initially, sodium borohydride (the reducing agent) was dissolved into ethylene glycol (72 mL) containing distilled water. An iron chloride hexahydrate (3 g) precursor was mixed into this solution under vigorous stirring followed by the addition of polyethylene glycol (2 g) and sodium acetate (6 g). The resultant reaction mixture was poured into a Teflon autoclave, kept in an oven at 200 °C for 10 h and cooled down in ambient conditions. Magnetic iron oxide nanoparticles were retrieved out of the solution via a magnet, rinsed by distilled water multiple times until the pH was neutralized and freeze dried. The mechanism and synthesis of iron oxide nanoparticles are exhibited in Figure 1a,b, respectively.

$$2FeCl_3 + 6NaBH_4 + 18H_2O \rightarrow 2Fe^0 + 6NaCl + 6B(OH)_3 + 21H_2$$

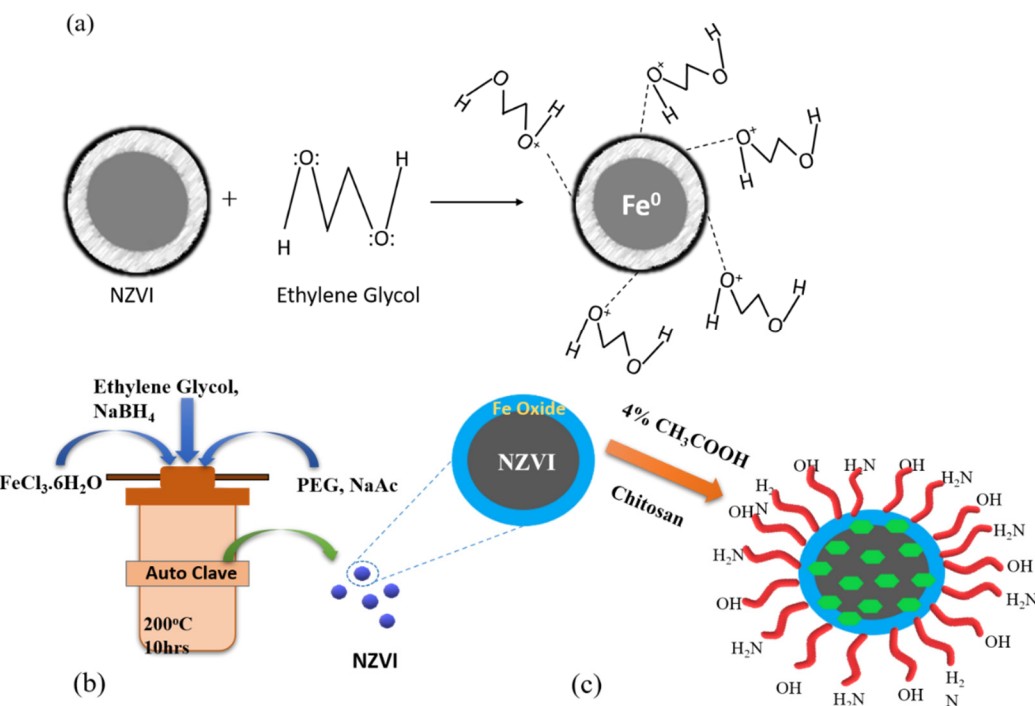

**Figure 1.** (**a**) Chemical equation and mechanism of nano zero valent iron oxide particles (NZVI) synthesis; (**b**) Scheme of NZVI synthesis; (**c**) Functionalization of NZVI by chitosan.

### 2.3. Synthesis of Iron Oxide Impregnated Chitosan

For preparation of hybrid beads, the template method was chosen [5]. Chitosan powder was dissolved into 4% acetic acid solution under continuous stirring overnight. Afterwards, various quantities of iron oxide nanoparticles were incorporated into the chitosan solution with vigorous stirring until the homogenous solution was obtained. The

compositions of the different samples were as follows: FeCh-20 contained 1 g (20%) iron oxide NPs and 4 g (80%) chitosan; FeCh-40 consisted of 2 g (40%) iron oxide NPs and 3 g (60%) chitosan; and FeCh-60 comprised 3 g (60%) iron oxide and 2 g (40%) chitosan. The resultant slurry was added dropwise into a 2M sodium hydroxide solution with the help of a syringe and stirred continuously until macro-sized spheres appeared. These macro spheres were separated using filter paper, rinsed using distilled water until pH 7 was obtained and dehydrated in a heating oven at 60 °C for 6 h. The experimental procedures are demonstrated in Figure 1c.

## 3. Characterization Techniques

An X-ray diffractometer (STOE, Darmstadt, Germany) was employed to study the crystalline structure of NZVI- and NPs-impregnated chitosan beads before and after adsorption of arsenic. Samples were scanned at 0.5 s/step in the 2θ range 10–90° and the Debye –Scherrer equation was utilized to calculate crystallite size. Infrared absorptions were measured in a 4000–400 cm$^{-1}$ wavenumber range by a Nicolet-6700 spectrophotometer by ThermoFisher Scientific, Massachusetts, United States to investigate the functional moieties present in NZVI and hybrid beads. The surface morphology of prepared samples was also studied using a JEOL-JSM-6490LA (JEOL, Tokyo, Japan) scanning electron microscope. Porosity percentage was calculated via Archimedes' rule by using ethanol as a displacement fluid, while pore size was determined via the BET technique. In compliance with the protocols of Archimedes' rule, the average porosity of the hybrid beads of various samples was estimated by using the following equation [22]

$$\in = \frac{V_s}{V_p + V_s}$$ (1)

where Vs is the volume of specimen (mL), Vp is the volume of pores (mL), and $\in$ shows the porosity of specimen.

The contact angle was measured by employing a sessile drop method at standard conditions. To restrict the spreading of each droplet's shape, contact angle values were recorded within 15–20 s after placing a drop on the surface of the beads. An atomic absorption spectrometer (AAS Vario 6, Analytik Jena, Germany) and an atomizer with a graphite furnace were used to analyze the concentration of arsenic in the samples. All analyses were performed using integrated absorbance. Argon was used as a purge gas for the analysis due to its inertness. The arsenic adsorption experiment was executed by applying the batch equilibrium technique. To study adsorption isotherm, 10–40 ppm initial arsenic concentrations were employed at neutral pH (6.9 ± 0.5) with a constant dose of hybrid beads (1 g/L) in an Erlenmeyer flask under continuous stirring at 150 rpm. Adsorption of As (V) was also studied as a function of pH by keeping the concentration of As (10 ppm) and the adsorbent (1 g/L) constant, while varying pH. After the adsorption experiment, the remaining concentration of arsenic was measured by the AAS and various theorems were employed to analyze the experimental results. The adsorption efficiency of the prepared materials was calculated using the following equations:

$$\%\text{age Removal} = \left[ \frac{(C_o - C_e)}{(C_o)} \right] \times 100$$ (2)

$$(q_e) = \frac{[(C_o - C_e) \times V]}{(m)}$$ (3)

$$\frac{C_e}{q_e} = \frac{1}{K_L \times q_m} + \frac{C_e}{q_m}$$ (4)

$$\log(q) = \log(K_f) + \frac{1}{n} \log(C_e)$$ (5)

where "$q_e$" is the adsorption capacity of beads (mg/g), "V" is the volume of solution (L), "m" is the mass of adsorbent (g), and "$C_o$" and "$C_e$" are the initial and final arsenic concentrations (mg/L), respectively. The process of adsorption has also been expressed by Langmuir and Freundlich relations (Equations (3) and (4)). The logarithmic form of the Freundlich equation is expressed in Equation (5), where "$K_f$" having unit (mg/g) is the Freundlich constant and "n" is the Freundlich exponent.

$$\ln(q_e - q_t) = \ln(q_e) - K_1 t \tag{6}$$

$$\frac{t}{q_t} = \frac{1}{K_2 q_e^2} + \frac{1}{q_e} t \tag{7}$$

To explore adsorption kinetics of arsenic on the hybrid beads, pseudo first order and pseudo second order kinetics (Equations (6) and (7)) were also applied, whereas Equation (8) was used to investigate the maximum adsorption.

$$q_e = \left( \frac{C_i - C_e}{W} \right) V \tag{8}$$

where "$q_e$" is equilibrium adsorption of arsenic, "Ci" and "Ce" are initial and equilibrium concentration (mg/L), respectively, "V" is the volume of solution (L), and "W" is the weight of dry adsorbent (g).

## 4. Results and Discussion

### 4.1. Structural Analysis of Virgin NZVI and Hybrid Beads

XRD was employed to measure crystallite size and to study the crystal architecture of prepared iron oxide NPs (Figure 2). The lattice spacing was quantified from 2θ reflection of peaks at 30°, 35.1°, 44.1°, 53.4°, 56.9° and 62.5°, which corresponded to planes 220, 311, 400, 422, 511 and 440, respectively [23]. The peak at 35.1° could be attributed to the oxidation of trace NZVI during preparation [24]. The absence of an extra peak confirmed the purity of the iron oxide while the solvent (water, ethanol) did not exhibit an effect on the crystal structure of the NPs. The peak pattern of NZVI matched well with database JCPDS file 01-089-0687 for iron oxide nanoparticles [25]. The XRD diffractogram also provided physical and chemical information about iron oxide NPs impregnated into chitosan beads. The diffraction peaks were observed at 2θ values of 20°, 35.1°, 43° and 62° [26]. These diffraction peaks were reconcilable with database JCPDS file 039-1894, with iron oxide particles having a spinel structure [27]. This confirmed the successful incorporation of iron oxide NPs in chitosan beads. The XRD diffractogram was used for the phase study of the adsorbed material after the arsenic adsorption experiment (Figure 2). Some peaks were detected at 2θ values of 11° and 35°, which indicates the adsorption of arsenic in crystalline form [28]. This is in agreement with arsenic XRD values in the database JCPDS file 00-008-0667. These results have confirmed that arsenic was successfully adsorbed onto iron oxide-embedded chitosan beads [29].

### 4.2. Functional Group Studies

IR spectra of NZVI are shown in Figure 3. The typical wide absorption signal at $531 \text{cm}^{-1}$ is due to vibrational stretching of the iron-oxygen functional moiety, while its shape and intensity indicate the crystalline structure of the magnetite property [30]. The characteristic absorbance at $1041 \text{ cm}^{-1}$ is attributed to the (C=O) vibration, while the signal at $1415 \text{ cm}^{-1}$ is due to the well-ordered vibrational stretching of the unidentate group of carbonates [31]. This can be attributed to the use of the ethylene glycol solvent with minor amounts of polyethylene glycol and sodium acetate. The absorption signals at $1555 \text{ cm}^{-1}$ and $1537 \text{ cm}^{-1}$ were attributed to the vibrational band of COO-Fe and the symmetric stretched band of -COO- bonds, respectively [32,33].

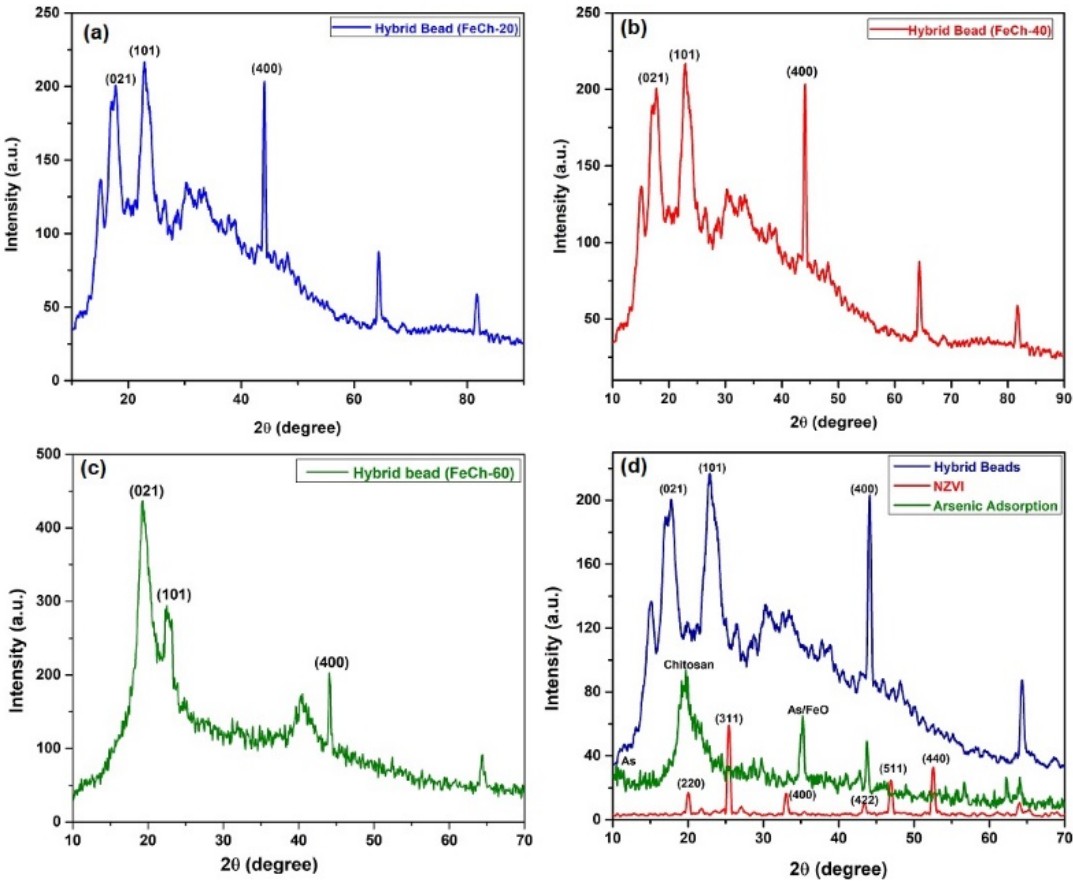

**Figure 2.** (**a**) XRD of FeCh-20 hybrid beads, (**b**) XRD of FeCh-40 hybrid beads sample, (**c**) XRD of FeCh-60 hybrid beads sample, (**d**) XRD of NZVI, NPs impregnated chitosan hybrid beads and hybrid beads after arsenic adsorption.

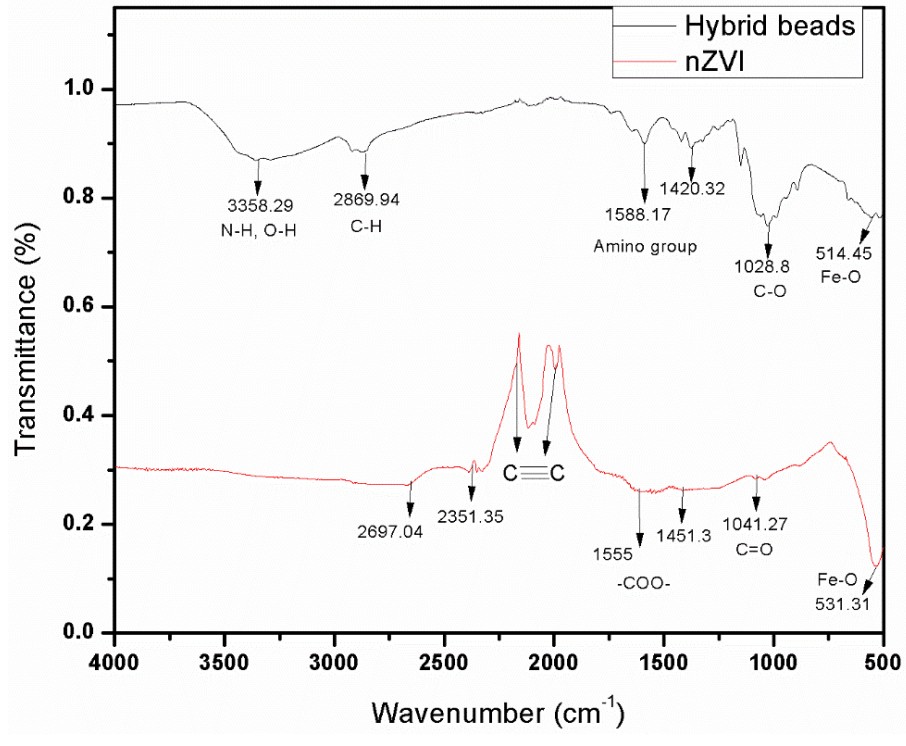

**Figure 3.** FTIR of NZVI and NPs impregnated chitosan hybrid beads.

IR spectra of exhausted NZVI-chitosan beads were recorded to understand the mechanism of arsenic adsorption (Figure 3). The characteristic band at 893 cm$^{-1}$ corresponded to the Fe-OH vibration of the akageneite (chloride containing iron hydro oxide) hybrid solution [34,35]. The absorption signal at 1375 cm$^{-1}$ was assigned to the C-O of the alcoholic group in chitosan [36], while absorption bands at 1028 cm$^{-1}$ and 1149 cm$^{-1}$ corresponded to the C-O single bond stretching vibrations in COC and CH$_2$OH categories of the pyranose ring. The absorption at 1588 cm$^{-1}$ was assigned to the amino group (N-H) stretching band. This signal was displaced from 1555 cm$^{-1}$ to 1588 cm$^{-1}$ after impregnation into chitosan due to the coordination effect [37]. The broad signal at 2869 cm$^{-1}$ was attributed to the C-H vibration stretching of methylene (-CH$_2$) and methyl (-CH$_3$) groups present in chitosan [35,38]. The vibrational signals marked at 3358 cm$^{-1}$ were due to N-H and O-H vibrational stretching in hybrid material [35,39]. The absorbance at 514 cm$^{-1}$ and 559 cm$^{-1}$ was related to the presence of the Fe-O group, attributed to the embedment of iron oxide particles [40].

### 4.3. Morphological Analysis of NZVI and Hybrid Beads

The SEM micrograph of NZVI (Figure 4a) exhibits a homogenous and regular distribution of iron oxide NPs with an average size of 150 nm. The morphology of NPs seems to be an agglomerate of small particles [41,42]. This may be due to a steric effect caused by the interaction of active points on NPs' surface and the magnetic interactions created by single particles [43].The surface analysis of the prepared hybrid beads indicates that NPs were successfully impregnated onto the chitosan. The SEM micrographs show that most of the NPs were embedded onto the chitosan matrix surface [44]. This configuration offers dual benefits of availability of NPs for adsorption and stability of bead shape that facilitates the handling of adsorbent materials in the column bed. SEM results (Figure 4b–d) of different samples show that the density of iron oxide NPs increased on the bead surface with the increased concentration of NPs during preparation [45].

Studying the surface of iron oxide-impregnated chitosan beads after the arsenic adsorption experiment was also performed using the SEM technique (Figure 4e,f). Distinctive changes were observed in the morphology of beads after adsorption. A multilayered crystal structure of arsenic was found on the surface of the hybrid beads, showing the good adsorption capacity of prepared materials for arsenic pollutants [46].

### 4.4. Porosity Studies of Hybrid Beads

The average pore size and pore diameter (P/P$_o$) of hybrid beads were measured using the BET technique based on the Barret–Joyner–Halenda (BJH) method (Figure 5). Average pore size and volume in hybrid beads were calculated using Equation (1), giving values of 4.55μm and 1.09 cm$^3$/g, respectively. The substantial pore size present on the surface of the hybrid beads may have assisted the mass transfer between the beads and arsenic [47]. In compliance with the protocols of Archimedes' rule, the average porosity of the hybrid beads from various samples was estimated as 67.16% using following equation [22].

### 4.5. Surface Wettability Analysis

The hydrophilic nature of a bead's surface increases its adsorption properties by improving wettability, adhesiveness and high surface energy [48]. The hydrophilic nature of the prepared hybrid beads was determined by contact angle analysis with water. The graph shown in Figure 6 reveals that the sample FeCh-60 possessed a hydrophobic nature ($\theta$ = 85°) due to a relatively lower amount of chitosan, whereas FeCh-20 and FeCh-40 demonstrated maximum surface wettability by presenting contact angles of 53° and 66°, respectively. Since chitosan is a hydrophilic material owing to -NH$_2$ and -OH groups, the hydrophilic nature of the hybrid beads was reduced by decreasing the chitosan concentration [49]. Moreover, it is associated with an increase in iron oxide NPs; a greater area of its surface would be occupied by NPs, which would reduce surface wettability. The wetting property

increased the contact area between the water and the surface of the hybrid beads which, in turn, enhanced the adsorption of arsenic.

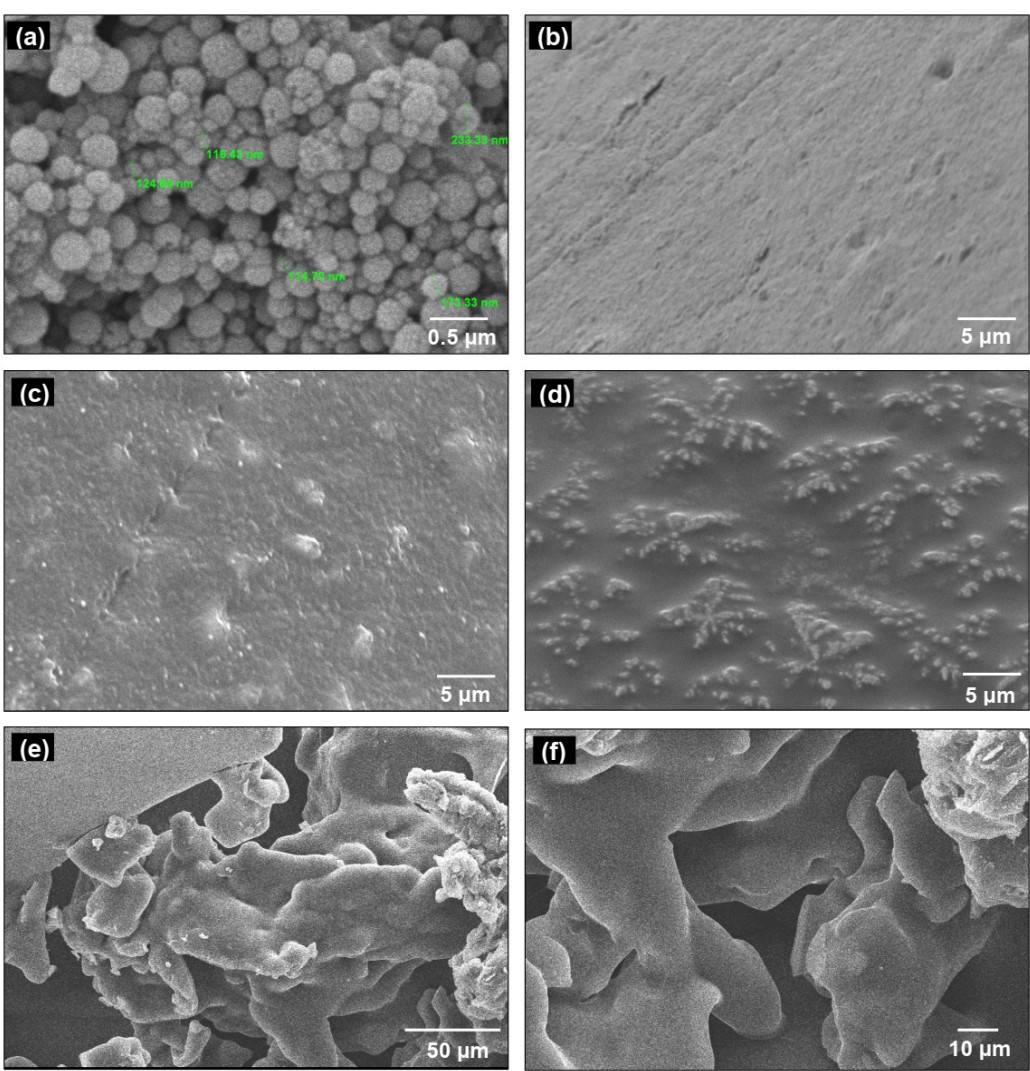

**Figure 4.** SEM micro images of: (**a**) NZVI, (**b**) hybrid bead FeCh-20, (**c**) hybrid bead (FeCh-40), (**d**) hybrid bead (FeCh-60), (**e**) hybrid beads FeCh-20 after arsenic adsorption, and (**f**) hybrid beads FeCh-40 after arsenic adsorption.

*4.6. Adsorption Efficiency*

An arsenic adsorption mechanism is proposed in Figure 7. The arsenic adsorption capacity of hybrid beads was investigated under varying parameters such as initial concentration of adsorbate, pH and contact time by applying different isotherm models, as shown in Figure 8. The ion exchange method was a very effective technique for arsenic removal, particularly when ferrous materials were employed as they were able to accommodate arsenic ions in interlayer voids, thus enhancing performance [50]. Initially, the As-O bond broke up via reduction upon adsorption onto the NZVI surface because the transfer of electrons was thermodynamically favored. This reduced arsenic diffusion from the oxide shell interface to the interior of the particle. Reducing agents such as $e^-$, $H_2$ or $Fe^{2+}$ were generated from NZVI corrosion in aqueous media. These penetrated through the magnetite layer to attack the arsenic, which formed an Fe-As bond via reduction. According to the literature [51], this may be due to an electron-transfer from Fe to As, thus producing an Fe-As bond and coagulate. At neutral pH, arsenic mainly exists in the form of $H_2AsO_4^-$ ions. The NZVI nanoparticles embedded within the complex structure of chitosan enhanced its stability and offered extra adsorbing active points for the arsenic [52]. The free amine

group present in chitosan could then be protonated in aqueous media. $AsO_4^{-3}$ anions were mostly bonded to amino groups of chitosan, which may form $R-NH_3^+$ after protonation with arsenate ions. The adsorption capacity of prepared materials is represented in Figure 8a. The highest adsorption capacity was measured at 18 mg/g for FeCh-40 owing to its maximum porosity and the hydrophilic property of its beads. Adsorption capacities of 14 mg/g and 16.7 mg/g were measured for FeCh-20 and FeCh-60, respectively. Moreover, FeCh-40 demonstrated the maximum removal of arsenic at 81%, as determined by Equation (2). The arsenic adsorption capacity ($q_e$) was also estimated by another relation given in Equation (3).

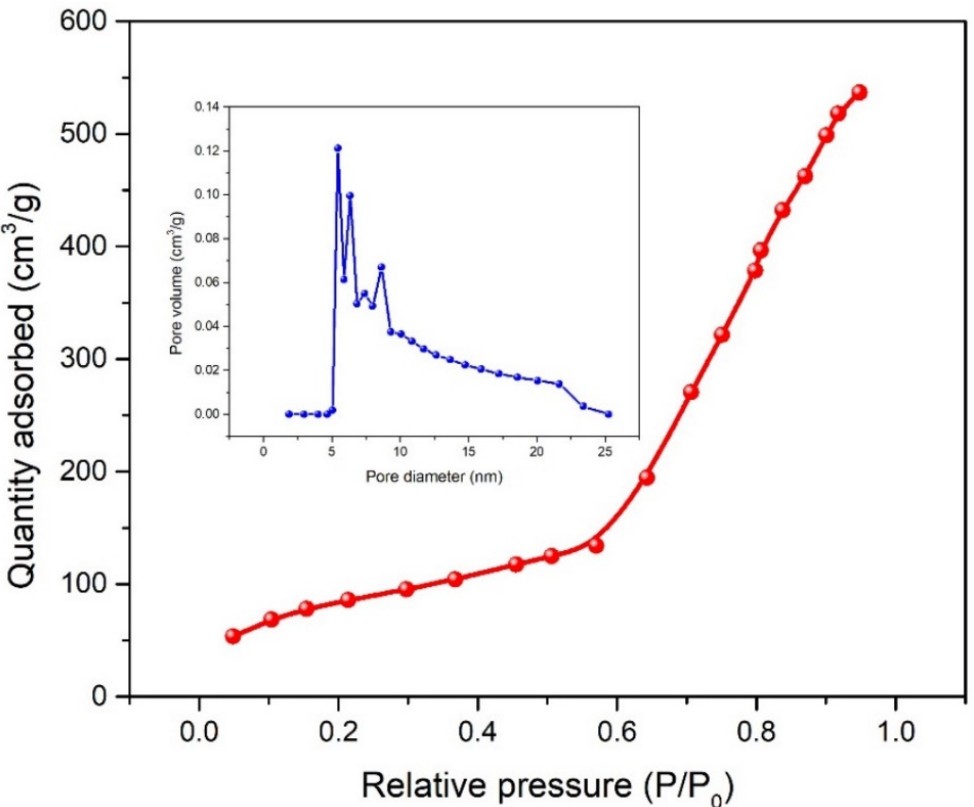

**Figure 5.** $N_2$ adsorption–desorption isotherms and BJH pore diameter distribution in hybrid beads.

### 4.6.1. Effect of Arsenic Initial Concentration

The batch processes for the treatment of arsenic in water were performed to estimate the arsenic adsorption capability of the hybrid beads, i.e., FeCh-20, FeCh-40 and FeCh-60. An arsenic solution was treated with 1.0 g of hybrid beads, and the arsenic initial concentration was varied within a 10–40 ppm range in an Erlenmeyer flask with continuous shaking at 150 rpm. Samples were evacuated at a constant time period of 10 h, while a neutral pH value was maintained to study the adsorption isotherm (Figure 8). The hybrid material FeCh-40 produced the best arsenic adsorption capacity values in comparison to some other adsorbents reported in the literature (Table 1). From these results, it is evident that the prepared hybrid materials performed excellently to remove arsenic contamination from water, achieving the standard set by WHO (10μg/L).

The process of adsorption has also been expressed by Langmuir and Freundlich relations (Equations (3) and (4)) in Figure 8c,d. According to the Langmuir isotherm, the value of $Q_{max}$ is 4.564 mg/g and $K_L$ is 1.074 l/mg. The regression coefficient ($R^2$) values of the Langmuir and Freundlich isotherms were 0.9951 and 0.98, respectively. These values have demonstrated that the Langmuir isotherm is the best fitted and that the process of adsorption was chemisorption on a homogenous surface [53].

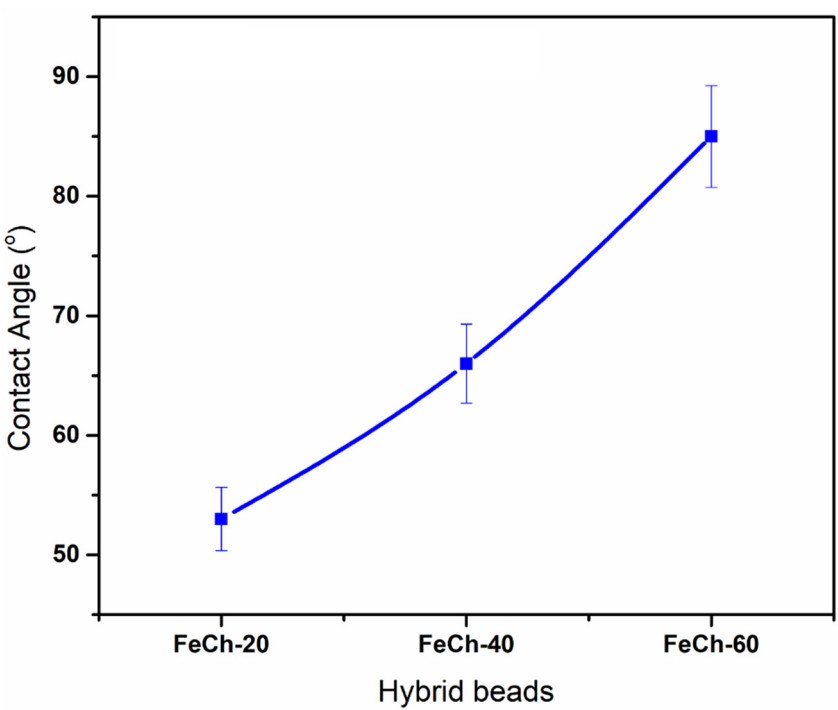

**Figure 6.** Contact angle of different hybrid beads samples: FeCh-20, FeCh-40 and FeCh-60.

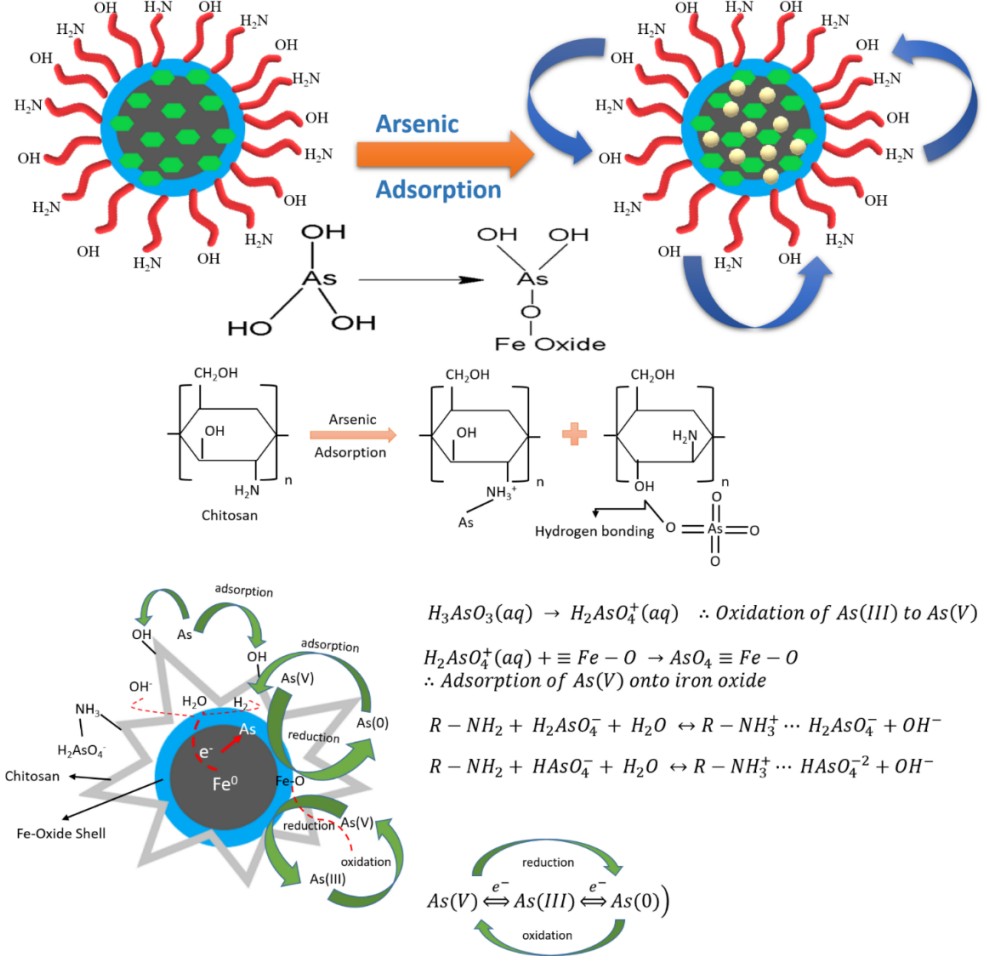

**Figure 7.** Illustration of arsenic adsorption scheme onto the surface of hybrid beads.

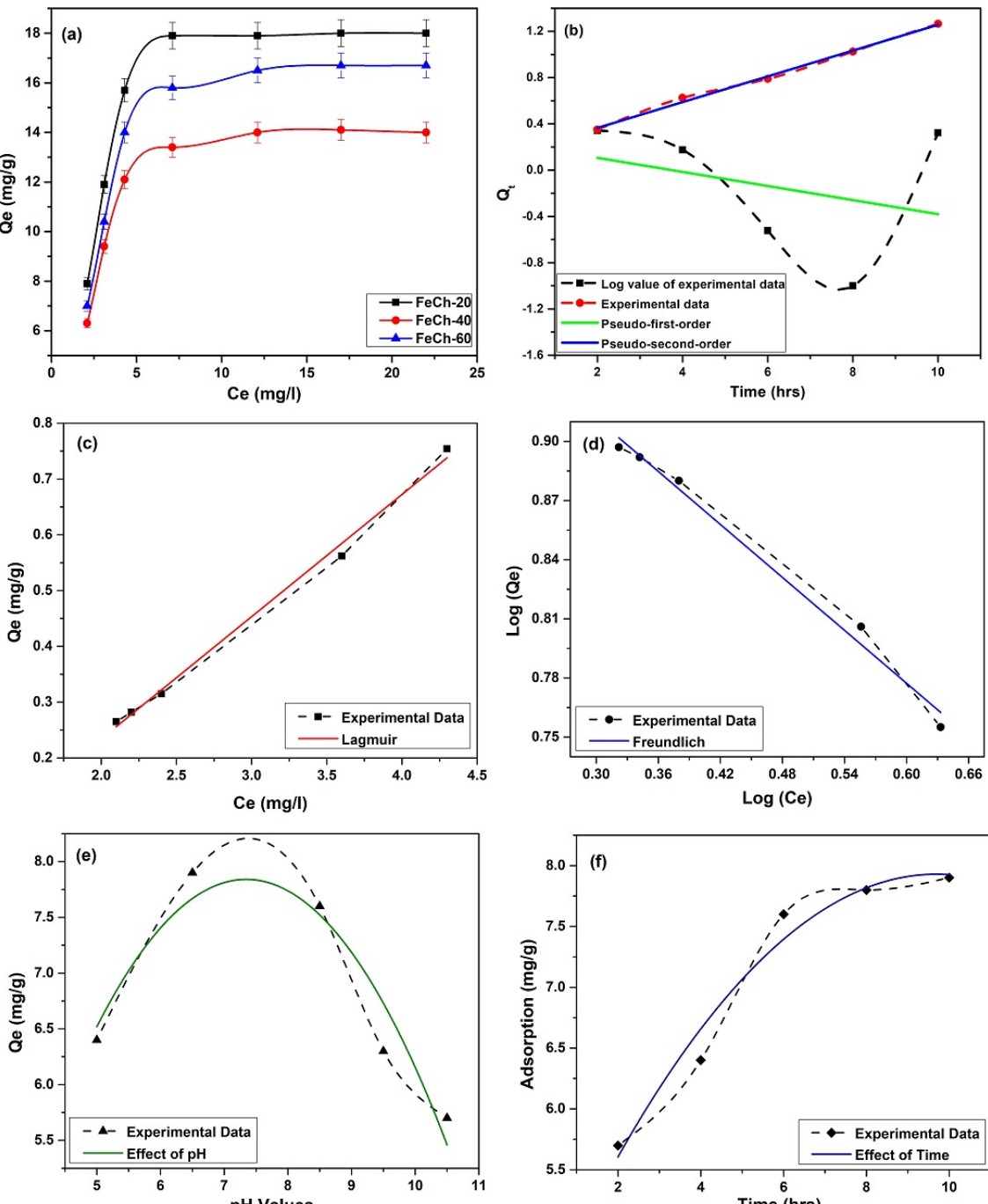

**Figure 8.** (**a**) Adsorption capacity of FeCh-20, FeCh-40 and FeCh-60 beads, (**b**) Pseudo first order and Pseudo second order kinetic model for arsenic adsorption on FeCH-40 beads, (**c**) Langmuir Model, (**d**) Freundlich Model, (**e**) Effect of pH value and (**f**) Effect of contact period.

**Table 1.** Comparison of adsorption values of prepared hybrid beads with other works reported in the literature.

| Adsorbent | Qe (mg/g) | pH | Reference |
|---|---|---|---|
| Hybrid NZVI-chitosan beads (FeCh-40) | 18 | 6.6–7.3 | Present work |
| Lateritic soil | 2 | 7 | [54] |
| Concrete/maghemite nanocomposites | 11.12 | 5 | [55] |
| Granulated iron hydroxide | 18 | 7 | [56] |
| Goethite modified (GT@DAS/TOES) | 7.78 | 6 | [55] |
| Chitosan goethite bio-nanocomposite (CGB) | 11.3 | 5–6 | [57] |
| Granular ferric hydroxide (GFH) | 8 | - | [58] |
| Iron oxide coated sponge | 4.5 | 6.5–7.3 | [59] |
| TiO$_2$-impregnated chitosan beads | 2.1 | 9.2–7.7 | [5] |

4.6.2. Adsorption Kinetics

In this study, a kinetic test of arsenic adsorption onto the surface of sorbent was also carried out for the adsorption rate measurement according to previous literature [7]. Figure 8b indicates the kinetics of arsenic removal from water with 1 g/L FeCh-40 sorbent under normal conditions. The kinetic curve has three major regions: initially, the adsorbate rapidly adsorbed onto the surface of the adsorbent; this gradual decreased until, finally, it became saturated when a constant mass was achieved due to maximum adsorption. To explore the adsorption kinetics of arsenic on the hybrid beads, pseudo first order and pseudo second order kinetics (Equations (6) and (7)) were also applied [60]. The value of regression ($R^2$) 0.995 shows that this adsorption process obeys the pseudo second order model. This predicts the performance over a complete range of studies, assists sustainability and supports the assumptions of the model that the process of adsorption occurred because of chemisorption [7].

4.6.3. Effect of TIME and pH on Arsenic Adsorption

The contact time required to attain an adsorption equilibrium between the arsenic solution and the hybrid beads was also investigated (Figure 8f). It is made evident by employment of Equation (8) that maximum adsorption ($q_e$) onto the surface of the beads was attained in 10 h. The initial adsorption rate was very fast because, in the beginning, there was a large number of active points on the surface as well as less resistance for mass transfer on the surface [61].

pH was a significant parameter that affected the adsorption properties of the beads' surface. The most appropriate pH condition was estimated as follows: by maintaining the pH of the arsenic solution during the adsorption experiment, while keeping the initial concentration of arsenic constant at 10 mg/L (Figure 8e). However, low pH conditions were not applied because chitosan polymer is unstable in acid environments [60]. According to results acquired by the use of Equation (8), the maximum arsenic adsorption on the beads' surface was measured within pH range 6–8. Because in aerobic water, arsenic exists in $H_3AsO_4$ form, it is dissociated into $HAsO_4^{2-}$ and $H_2AsO_4^{-}$ ionic species in aqueous solution in the pH range 6–8 [62]. This would be favorable for the process of adsorption due to electrostatic attraction, assisting ligand interchange and protonation on the surface of the adsorbent [48,63]. However, with the increase in the pH of the test solution, negatively charged sites on the surface of the sorbent also increased, which diminished the rate of adsorption because of electrostatic repulsion. Hence, these hybrid beads demonstrated a maximum performance of arsenic adsorption at neutral pH, which is highly applicable for the treatment of drinking water.

**5. Conclusions**

Novel hybrid beads of chitosan-functionalized zero valent iron oxide were synthesized by varying concentrations of chitosan and iron oxide. These prepared materials were characterized by FTIR, XRD, SEM and BET techniques. The study showed that a maximum of

81% removal of arsenic was achieved within 10 h at neutral pH under optimum conditions of adsorption by a FeCh-40 hybrid bead sample. The adsorption behavior was satisfactorily illustrated through Langmuir and Freundlich models, while the adsorption mechanism was studied by pseudo first order and pseudo second order kinetics relations. Equilibrium results have recognized the best fitted models as being the pseudo second order, with the correlation coefficient 0.995, and the Langmuir isotherm model, with regression coefficient 0.99, while homogenous and monolayer adsorption were identified as the prevailing adsorption mechanism. In this study, the highest arsenic adsorption capacity was achieved at 18 mg arsenic per g of FeCh-40 adsorbent, whereas 11.2 mg arsenic elimination was reported by maghemite nanocomposites in the literature.

**Author Contributions:** M.F.A. carried out the experimental work. N.M.A. is the principal investigator of the project and led the project overall. M.A.A. helped to analyze the adsorption data. A.M., A.U.K., R.S. and S.Z. edited the manuscript and supervised experimental work. N.A.K. and M.A.Q. assisted in the design of experiments. H.R. helped to perform the arsenic removal experiment and analyses. H.Q. helped in the formatting of the manuscript. All authors have read and agreed to the published version of the manuscript.

**Funding:** This research work has not received any external funding.

**Institutional Review Board Statement:** Not applicable.

**Informed Consent Statement:** Not applicable.

**Data Availability Statement:** All the data is provided in this article.

**Acknowledgments:** All authors are grateful to the NUST Research Directorate for financial support. The corresponding author has acknowledged funding from HEC-NRPU through Project No. 6020.

**Conflicts of Interest:** The authors declare no conflict of interest.

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
