# Peer review of "Hybrid Beads of Zero Valent Iron Oxide Nanoparticles and Chitosan for Removal of Arsenic in Contaminated Water"

_water, doi:10.3390/w13202876_

Round 1

Reviewer 1 Report

The manuscript of Ahmed et al. studies the removal of arsenic from contaminated water using iron nanoparticles (NP) functionalized with chitosan. The topic is exciting, and there is a great interest in developing these types of adsorption systems. In particular, the adsorption of Arsenic(V) is studied.

Iron NPs were synthesized by hydrothermal method and after that impregnated with chitosan. Three samples with different quantities of chitosan are explored. 

The techniques used are XRD to determine the structure of the NPs, FTIR to analyze the mechanism of arsenic adsorption, SEM images of the different materials with and without arsenic, and ICP to determine the adsorption capacity (Qe). In addition, the effect of pH and time is also analyzed.  

The most important result is that the adsorption system developed works well at pH values close to 7, which is the pH value of drinkable water.

I think the topic of the manuscript is exciting, and the result deserves publication, but the authors should improve several parts of the manuscript. Here, my comments:

1. The manuscript has several language mistakes. The authors should extensively revise the language of the entire manuscript. In addition, the number of the figures are wrong (see page 11, figure 1 should be figure 7; page 12, figure 2 should be Figure 8, etc.). 

2. Introduction need more reference on the use of different types of iron oxide for arsenic removal. Several recent papers are available like T. Mishra et al J. Environ Chem Engg, 4 (2016) 1224; Torasso et al J. Environ Chem Eng (2021).

3. In several places, the authors use the word “impregnated with chitosan” when referring to the beads developed in this work. My question is if the nanoparticles are impregnated or functionalized with chitosan. In addition, I would like to know if the adsorption system is a bead or if a matrix of chitosan is filled with iron oxide nanoparticles.

4. Adsorption experiments were carried out in broad pH values but, in particular, at low pH values (2-4) for several hours. What happens to the iron oxide nanoparticles at that pH? If H+ penetrates, the iron oxide will dissolve. In such a case, what is the reusability of the cleaning system?

5. Adsorption of arsenic from water need reusability for a long time for practical application. What is your comments on it as several materials were already developed in the literature? 

6. Batch adsorption procedure part was poorly written. The authors should mention the time and agitation/shaking speed for the isotherm experiment. The amount of adsorbent should be included. 

7. As As(III) species is more toxic than As(V), authors should include As(III) removal performance of the system.

8. Analysis of adsorption isotherm data with Langmuir model should be done correctly by providing the values of all the parameters. It must be compared with other isotherm models for a better understanding of the adsorption process.

9. Authors should evaluate the competitive effects of other common ions in the water towards As removal using this system.

10. In Figures 2 (c and d), the lines should be the model and the experimental data in points.

Author Response

Dear Reviewer

We added responded the comments in best possible hope. Hope you will be satisfied by the response

Reviewer 2 Report

1. Why the author chose to prepare the iron oxide nanoparticles yourself not buy them? As the prepared-self might not as the expected.
2.  The description of each symbol about equation (1) and equation (8) should be added.
3. The standard XRD pattern of each oxide should be added in Figure 2.
4. The description content in the section of "4.2. Functional Group studies" was not consistent with the Fig.3, please check and revised, especialy the wavenumbers of the IR.  What are the two peaks at nearly 2200 cm-1 for NZVI? why these two obvious peaks were not analyzed while other little peaks were analyzed? 
5. The Figure captions were confused, please revise them. The Figure 1 in section 4.6 should be Figure 7, and the Figure 2 should be Figure 8.
6. The section 4.7 was missed while 4.7.1, 4.7.2 and 4.7.3 were all in.

Author Response

Dear Reviewer

Attached the response to your comments . These comments help us to improve the manuscript and hope it will also be satisfactory for you.

Round 2

Reviewer 1 Report

The authors responded to all my questions and I think the manuscript can be published as-is. 

Author Response

Dear Reviewer,

We highly appreciate your suggestions that help to improve our manuscript.

Reviewer 2 Report

The author had improved their paper according to the reiewers' comments, it was recommended for publication after minor revision.

(1) Figure 7 and Figure 8 were not displayed in the paper.

(2) The Figure captions were confused, please revise them. The Figure 1 in section 4.6 should be Figure 7, and the Figure 2 should be Figure 8.

(3) Sub-figures in Figure 8 (Figure 2 in the paper)should be in same size and the resolution of (a) and (b) should be improved.

Author Response

(The authors gave the same response as above.)
